# RedquorinXS Mutants with Enhanced Calcium Sensitivity and Bioluminescence Output Efficiently Report Cellular and Neuronal Network Activities

**DOI:** 10.3390/ijms21217846

**Published:** 2020-10-22

**Authors:** Adil Bakayan, Sandrine Picaud, Natalia P. Malikova, Ludovic Tricoire, Bertrand Lambolez, Eugene S. Vysotski, Nadine Peyriéras

**Affiliations:** 1Institut de Neurobiologie Alfred Fessard, UPR 3294, Centre National de la Recherche Scientifique (CNRS), Avenue de la Terrasse, 91198 Gif-sur-Yvette, France; 2BioEmergences Unit, CNRS USR 3695, Université Paris-Saclay, Avenue de la Terrasse, 91198 Gif-sur-Yvette, France; 3Neuroscience Paris Seine—Institut de Biologie Paris Seine (NPS—IBPS), CNRS UMR8246, INSERM U1130, Sorbonne Université UM119, 75005 Paris, France; ludovic.tricoire@upmc.fr (L.T.); bertrand.lambolez@upmc.fr (B.L.); 4Photobiology Laboratory, Institute of Biophysics SB RAS, Federal Research Center “Krasnoyarsk Science Center SB RAS”, 660036 Krasnoyarsk, Russia; npmal@yandex.ru (N.P.M.); eugene_vysotski@ibp.ru (E.S.V.)

**Keywords:** bioluminescence, aequorin, calcium sensor, BRET, mutagenesis, GPCR assay, neuronal network imaging

## Abstract

Considerable efforts have been focused on shifting the wavelength of aequorin Ca^2+^-dependent blue bioluminescence through fusion with fluorescent proteins. This approach has notably yielded the widely used GFP-aequorin (GA) Ca^2+^ sensor emitting green light, and tdTomato-aequorin (Redquorin), whose bioluminescence is completely shifted to red, but whose Ca^2+^ sensitivity is low. In the present study, the screening of aequorin mutants generated at twenty-four amino acid positions in and around EF-hand Ca^2+^-binding domains resulted in the isolation of six aequorin single or double mutants (AequorinXS) in EF2, EF3, and C-terminal tail, which exhibited markedly higher Ca^2+^ sensitivity than wild-type aequorin in vitro. The corresponding Redquorin mutants all showed higher Ca^2+^ sensitivity than wild-type Redquorin, and four of them (RedquorinXS) matched the Ca^2+^ sensitivity of GA in vitro. RedquorinXS mutants exhibited unaltered thermostability and peak emission wavelengths. Upon stable expression in mammalian cell line, all RedquorinXS mutants reported the activation of the P2Y2 receptor by ATP with higher sensitivity and assay robustness than wt-Redquorin, and one, RedquorinXS-Q159T, outperformed GA. Finally, wide-field bioluminescence imaging in mouse neocortical slices showed that RedquorinXS-Q159T and GA similarly reported neuronal network activities elicited by the removal of extracellular Mg^2+^. Our results indicate that RedquorinXS-Q159T is a red light-emitting Ca^2+^ sensor suitable for the monitoring of intracellular signaling in a variety of applications in cells and tissues, and is a promising candidate for the transcranial monitoring of brain activities in living mice.

## 1. Introduction

Intracellular Ca^2+^ is a critical signal in many important physiological processes and pathophysiological conditions across living species, and measurements of intracellular Ca^2+^ concentration are widely used in virtually all domains of biology [1]. Ca^2+^-dependent photoproteins are a family of bioluminescent molecules that emit blue light upon Ca^2+^ binding and share high structural and functional homology [2,3]. These photoproteins contain three EF-hand Ca^2+^ binding sites and form a stable complex from the reaction of the apo-photoprotein (luciferase) with coelenterazine (CLZ, the luciferin; [2,3]). Light emission proceeds with a rate that increases with Ca^2+^ concentrations in the 0.1 µM to 1 mM range [4,5,6,7]. Ca^2+^-dependent photoproteins exhibit rapid Ca^2+^ binding/unbinding kinetics, low light output in the absence of Ca^2+^, and broad dynamic range [6,8,9]. Aequorin is the photoprotein of choice in Ca^2+^ sensing applications, owing to the wealth of information on its structure, function, and its selectivity for Ca^2+^ [3,10,11,12,13]. However, as for other Ca^2+^-dependent photoproteins, the bioluminescence from wild-type (wt) aequorin is emitted at a slow rate at low Ca^2+^ concentrations, the light emission intensity below 1 µM Ca^2+^ being less than one-thousandth of that obtained in 1 mM Ca^2+^ [6,9,14]. This low bioluminescence output significantly limits the spatiotemporal precision of aequorin as a reporter of intracellular Ca^2+^ signals, which are in the 0.1–0.5 µM range in most mammalian cell types.

Several approaches help in circumventing this low Ca^2+^ sensitivity. The incorporation of various CLZ analogs is a convenient way to alter Ca^2+^ sensitivity, decay kinetics, and the emission spectrum of aequorin [15,16,17,18,19]. Although CLZ analogs often suffer from low water solubility or cell membrane permeability, aequorin reconstituted with the CLZ-f analog has been used in high sensitivity Ca^2+^ measurements in cellulo or in vivo [20]. Alternatively, the use of apo-aequorin mutagenesis should, in principle, allow for altering aequorin Ca^2+^ sensitivity for applications where the CLZ analog choice is detrimental. However, while aequorin mutants with decreased Ca^2+^ sensitivity have been successfully applied to the study of cellular organelles containing high Ca^2+^ concentrations [21,22], only one aequorin mutant showing enhanced light output at low Ca^2+^ concentrations has been described to date [7,23]. Finally, the fusion of the green fluorescent protein (GFP) with aequorin (GA), which naturally interact in the jellyfish *Aequorea victoria* [2], results in over twenty fold enhancement of GA sensitivity to submicromolar Ca^2+^ concentrations, as compared to aequorin alone [24]. Likewise, GA is a dual fluorescent label and Ca^2+^ sensor emitting green light through Förster resonance energy transfer (FRET), which enables the imaging of Ca^2+^ signals in single cells, tissue slices, and whole animals [23,24,25,26,27,28,29,30].

Aequorin fusion with GFP yellow mutants, or with red fluorescent proteins, further shifts light emission to the red, thereby enhancing light output through biological tissues [19,31,32,33]. The tdTomato-aequorin fusion named Redquorin [33] is spectrally optimal for deep tissue in vivo imaging due to its large and almost complete red shift (peak at 582 nm). However, its low sensitivity to submicromolar Ca^2+^ concentrations limits its usefulness as a sensor of intracellular Ca^2+^ signals. In the present work, we aimed at enhancing Redquorin Ca^2+^ sensitivity through apo-aequorin mutagenesis. We first screened apo-aequorin molecules mutated at selected amino acid positions for their sensitivity to Ca^2+^. Apo-aequorin mutants showing extra sensitivity to Ca^2+^ (AequorinXS) were then used to generate six Redquorin mutants that were characterized in vitro and in cellulo. Four of these mutants, named RedquorinXS, were stably expressed in CHO cell lines and tested for their sensitivity to physiological Ca^2+^ signals. Finally, a RedquorinXS Ca^2+^ sensor was expressed in a mouse brain slices through viral transfer and validated for the imaging of neuronal network activities.

## 2. Result

### 2.1. Identification of New Mutations That Enhance Aequorin Ca^2+^ Sensitivity

A total of twenty-four amino acid positions that belong to EF-hand domains and their surroundings in the 3D structure of apo-aequorin [10,11,12] were selected for mutagenesis (Figure 1). Seven of these amino acids belong to the EF-hand consensus sequences of apo-aequorin, including four Ca^2+^ binding residues. The mutagenesis consisted of random and non-conservative amino acid conversions at single amino acid positions. The resulting mutants were screened for Ca^2+^ affinity change in vitro. Apo-aequorin mutants were transiently expressed in HEK cells, purified, and reconstituted with CLZ-f. A first screening revealed six mutations (Y38S; A69M; E84K; I107E; C180S; and G185N) that yielded very low or non-detectable luminescence at pCa 6 and were not further considered.

The analysis of Ca^2+^ titration curves of aequorin mutants resulted in the identification of five groups with different performances. A large group of aequorin mutants whose sensitivity to Ca^2+^ was not significantly altered (Figure 2A and Appendix A), and four groups with significant changes in Ca^2+^ sensitivity (Figure 2B, Table 1 and Appendix A) compared to wt-aequorin (wt-aeq). Two of these groups (AequorinXS mutants) contained mutations that resulted in medium-high to high sensitivity to Ca^2+^ (e.g., A123D; A179T; Q159D; and Q159T). Interestingly, different amino acid substitutions at the same positions resulted in different, sometimes opposite changes in Ca^2+^ sensitivity (e.g., Q159K vs. Q159D, S157D vs. S157A, A179T vs. A179E, Figure 2B), thereby highlighting the importance of these specific amino acid residues in aequorin bioluminescence.

EC50 values derived from the fit of Ca^2+^ titration curves (Table 1, upper part) showed that, in comparison with wt-aequorin (659 nM), mutations Q159T (310 nM) and Q159D (330 nM) largely increased Ca^2+^ sensitivity, as well as S157D (322 nM), A179T (330 nM), and A123D (374 nM). The examination of light emission intensities of these mutants at pCa 6.5 and pCa 7.2 (relative intensity, Table 1, upper part) confirmed their enhanced Ca^2+^ sensitivity relative to wt-aequorin. Mutants Q159D and Q159T exhibited the highest luminescence output at pCa 6.5, their light emission intensities being 25 and 16 time higher than wt-aequorin, respectively. Moreover, the Q159D mutant showed a drastically higher (23-fold) luminescence output than wt-aequorin at the minute Ca^2+^ dose of pCa 7.2, suggestive of its potential usefulness at detecting low Ca^2+^ concentrations.

A previous study has reported that mutations of the first EF-hand domain can increase aequorin Ca^2+^ sensitivity [7]. However, these latter mutants exhibit slower light emission rate than wt-aequorin, which translates into slower decay kinetics of light emission in the continuous presence of Ca^2+^ [7]. Because slow rate/slow decay of light emission can hamper the use of aequorin mutants for monitoring fast intracellular Ca^2+^ dynamics, we compared the bioluminescence half decay time (*t_1/2_*) of aequorin mutants identified herein with that of wt-aequorin, in saturating Ca^2+^ concentration. The *t_1/2_* values of AequorinXS mutants (Table 1, upper part) were in the range of wt-aequorin *t_1/2_* (906 ms), with Q159D and A179T mutants showing significantly faster decay kinetics (*t_1/2_*: 794 and 779 ms, respectively) than wt-aequorin.

Because amino acid mutations can shift the wavelength of aequorin luminescence [34], we established the light emission spectra of the AequorinXS mutants. None of these mutants exhibited significant change of luminescence spectrum peak relative to wt-aequorin (Table 1, upper part). This suggests that the present apo-aequorin mutations enhanced the Ca^2+^ sensitivity of aequorin without notably affecting the apo-aequorin-coelenterazine reaction per se.

We next reasoned that combining some of the above single mutations may further enhance the Ca^2+^ sensitivity and light emission rate of aequorin. Indeed, Ca^2+^ titration analyses of aequorin double mutants (associating Q159D or Q159T with N121D, A123D, and A179T) revealed that all these combinations resulted in further enhanced Ca^2+^ sensitivity and light emission rate (Table 1 lower part, and Figure 3). Among double mutants, the Q159D+A179T and Q159T+A123D mutants exhibited the highest Ca^2+^ sensitivity (QD+AT and QT+AD, *EC50*: 216 nM and 279 nM, respectively). Moreover, the QD+AT and QT+AD mutants additionally showed the fastest light emission rate (*t_1/2_*: 612 ms and 620 ms, respectively). The Q159D, Q159T, QD+AT, QD+AD, QT+AT, and QT+AD aequorin mutants, termed Aequorin XS, were thus selected as the most promising candidates for enhancing the Ca^2+^ sensitivity of aequorin-based Ca^2+^ sensors.

### 2.2. Generating Redquorin Mutants with Ca^2+^ Sensitivity and Light Emission Rate That Match Those of GA

The fusion of aequorin with GFP (GA) shifts the light emission to green and enhances Ca^2+^ sensitivity [24], whereas its fusion with tdTomato (Redquorin) emits red light, thus allowing deep tissue imaging, but does not enhance Ca^2+^ sensitivity [32,33]. In order to improve Redquorin as a reporter of intracellular Ca^2+^ signals, we incorporated the above Q159D, Q159T, QD+AT, QD+AD, QT+AT, and QT+AD mutations into its apo-aequorin moiety, and compared the resulting mutants with Redquorin and GA, and with the fusion of wt-aequorin with the citrine yellow variant of GFP (CitA, [33]). Aequorin moieties of fusion proteins were reconstituted with native CLZ or CLZ-f and their Ca^2+^ sensitivities, light emission rates, and emission peak wavelengths were determined (Figure 4 and Table 2).

The Ca^2+^ sensitivity of CitA, which has not been previously characterized in detail, was compared with those of Redquorin and GA (Figure 4 and Table 2). Their overall Ca^2+^ sensitivities followed the order Redquorin < GA < CitA, as indicated by their EC50 values (CitA<GA<Redquorin), light emission intensities at pCa 6.5 and 7.2 (Redquorin<GA<CitA), and light emission rates at saturating Ca^2+^ concentration (decay *t_1/2_*: CitA<GA<Redquorin). All Redquorin mutants showed increased Ca^2+^ sensitivity relative to Redquorin, as judged from their lower Ca^2+^ EC50 and higher light emission intensity at both pCa 6.5 and 7.2, regardless of the CLZ used (Figure 4 and Table 2). The light emission rate of the mutants at saturating Ca^2+^ concentration was also higher (i.e., smaller *t_1/2_*) than that of Redquorin in both CLZ conditions, whereas the emission peak wavelength was unchanged (Table 2). All Redquorin mutants, except RedquorinQ159D in native CLZ condition, showed higher sensitivity to low Ca^2+^ than GA, as indicated by their lower Ca^2+^ EC50 and higher light emission intensity at both pCa 6.5 and 7.2 (Figure 4 and Table 2). However, the light emission rate of all mutants at high Ca^2+^ concentration was smaller (i.e., larger *t_1/2_*) than that of GA, except RedquorinQT+AD in CLZ-f condition (Table 2). Redquorin mutants had lower Ca^2+^ sensitivity than CitA, except RedquorinQD+AT that exhibited higher sensitivity to low Ca^2+^ but lower light emission rate at high Ca^2+^ concentration in both CLZ conditions, except RedquorinQT+AD that showed higher light emission rate at high Ca^2+^ concentration in CLZ-f condition (Figure 4 and Table 2). Interestingly, the use of CLZ-f instead of native CLZ differentially enhanced the overall Ca^2+^ sensitivity of the present aequorin fusion molecules, this effect being more pronounced for CitA and most Redquorin mutants (including RedquorinQ159D), than for GA, Redquorin and RedquorinQ159T (Figure 4 and Table 2). Hence, AequorinXS mutations enhanced all aspects of Redquorin Ca^2+^ sensitivity, with some Redquorin mutants exhibiting CitA-like properties (e.g., RedquorinQD+AT for sensitivity to low Ca^2+^, and RedquorinQT+AD for light emission rate at high Ca^2+^ in CLZ-f condition), and others being close to GA (e.g., RedquorinQ159T, notably in CLZ-f condition). These Redquorin mutants are thus promising candidate Ca^2+^ sensors for a variety of applications in cellulo, in tissue slices, and in vivo.

### 2.3. Effects of Key Amino Acid Mutations on Redquorin Thermostability

Aequorin-based biosensors have been employed to report intracellular Ca^2+^ signals at body temperature in mice [26,32]. However, it is documented that some apo-aequorin point mutations can decrease aequorin thermostability [35]. We thus compared the thermostability of the present aequorin and Redquorin mutants with those of wt-aequorin, Redquorin, GA, and CitA by incubating purified photoproteins for 30 min at 30 °C or 40°C. While some mutations slightly reduced aequorin thermostability (up to 29% decrease relative to wt-aequorin, see S157D at 40 °C), most aequorin and Redquorin mutants showed little or no decrease of activity compared to wt-aequorin, Redquorin, GA and CitA after incubation at 30 °C or 40 °C (Figure 5 and Appendix A). These findings indicate that the present Redquorin mutants are suitable for monitoring intracellular Ca^2+^ changes with high sensitivity at the body temperature in mammals.

### 2.4. RedquorinXS Mutants Efficiently Report the Activation of P2Y2 Receptor in CHO Cells

Ca^2+^-activated photoproteins, such as aequorin, have been long used for the detection of intracellular Ca^2+^ changes triggered by activation of G-protein-coupled receptors (GPCRs) [36], and for the screening of drugs acting at these receptors [37]. Here, a set of four Redquorin mutants were tested for their ability to be converted into active photoproteins following stable expression in Chinese Hamster Ovary (CHO) cells. Their performance was assayed through the activation of endogenous P2Y2 purinergic receptors by their natural agonist ATP. All Redquorin mutants showed similar cytoplasmic and homogeneous expression pattern (data not shown). The assay for the activation of endogenous P2Y2 receptor was performed with CHO cells stably expressing GA, Redquorin, Redq/Q159T, Redq/Q159D, Redq/QD+AT or Redq/QT+AD. In CHO cells, the activation of P2Y2 receptors by external application of ATP leads to activation of the IP3 (inositol 1, 4, 5-triphopsphate) signaling pathway and an increase in the cytoplasmic Ca^2+^ concentration. Upon P2Y2 receptor activation by ATP, photoprotein-expressing CHO cells showed a prominent, dose-dependent bioluminescent response, specific for each of these Ca^2+^ sensors (Figure 6). A detailed analysis of the ATP response curve is presented in Table 3.

The values of half-maximal effective concentrations (*EC50*) for P2Y2 activation determined from the fit of dose-response curves were around 2 µM for most CHO cell lines expressing Redquorin mutants or GA (Figure 5 and Table 3), in accordance with earlier reports [13,18]. The expression of these latter Ca^2+^ sensors also resulted in submicromolar ATP concentration detection limits (Table 3). In particular, the cell line Redq/Q159T gave an EC50 value of 1.7 µM, which closely matched that of the GA-expressing cell line (1.5 µM), and exhibited the lowest ATP concentration limit of all Ca^2+^ sensors tested (310 nM, Table 3). In contrast, the cell lines expressing native Redquorin and the Redq/QT+AD mutant, yielded higher EC50 values (4.0 and 3.0 µM, respectively), associated with supramicromolar ATP concentration limits (Table 3).

In high-throughput screening assays, the number of compounds is evaluated, and thus the assays need to be robust and reproducible over time. Hence, the need for at least one strict assay validation parameter that assures high quality data and suitability of the system. For validating this Ca^2+^ sensor-based assay, ATP experiments were performed at the maximum and minimum signal or response levels in order to ensure that the signal window is adequate to detect effective ATP concentrations. Accordingly, the Z-factor was calculated for each pair of CHO cell line/ATP response, which reflects the assay signal dynamic range and the data variation associated with the signal measurements [38]. Values of ≥ 0.6 are commonly considered to indicate a valuable assay (which is comparable to a signal window ≥3). The data showed that the cell line CHO/Redq/QT+AD performed best in terms of signal measurement (Z-factor = 0.82), although this cell line lacked submicromolar sensitivity to ATP, whereas the performance of the cell line expressing native Redquorin was the lowest (Z-factor = 0.56). The cell line CHO/Redq/Q159T exhibited an excellent Z-factor value of 0.76 and performed well in responding to submicromolar ATP concentration (Table 3). Hence, all Redquorin mutants performed better than native Redquorin in terms of sensitivity to physiological Ca^2+^ signals, and GPCR assay robustness and reproducibility.

### 2.5. Imaging Neuronal Network Activities in Brain Slice with a RedquorinXS Mutant

GA has been used for the real-time monitoring of intracellular Ca^2+^ increase associated with bioelectrical activity in single neurons or multiple neurons simultaneously in brain slices [23,30]. In order to evaluate the usefulness of RedquorinXS mutants as neuronal activity sensors, we compared the performance of GA with that of RedquorinQ159T (whose Ca^2+^ sensing properties are closest) for the imaging of neuronal network activity in slices of mouse neocortex. After the overnight incubation of acute cortical slices with recombinant Sindbis virus encoding either GA or RedquorinQ159T, the expression of sensors was visualized using GFP or tdTomato fluorescence, respectively. Sensor-expressing cells were labeled throughout the somatodendritic compartment, localized primarily in layers II/III and V, and exhibited the morphology of excitatory pyramidal neurons (Figure 7), consistent with earlier observations on GA [23,30]. A large majority of Redquorin-expressing cells (80.9% in layer II/III, *n*=110; 75.3% in layer V, *n* = 77) were immunoreactive for Satb2, which is selectively expressed in excitatory neurons of the mature cortex ([39], Figure 7). These results indicate that most sensor-expressing cells following Sindbis viral transfer were pyramidal neurons, as described for GA [30].

We compared the ability of the two sensors to report neocortical network activities elicited by the removal of extracellular Mg^2+^ (*n* = 2 slices and 14 slices for GA and RedquorinQ159T, respectively), which is known to induce episodes of synchronized epileptiform activity of the network that can be monitored using GA bioluminescence [30]. In the control conditions, the whole-field recording basal signal was low (below 10^3^ cps for both sensors, Figure 7), consistent with the low spontaneous activity observed in neocortical slices [40] and the low background luminescence of aequorin in absence of Ca^2+^ [6]. After removal of extracellular Mg^2+^, bioluminescence peaks of large amplitude (reaching above 10^5^ cps for both sensors) gradually became more frequent on the traces of whole field recording, reflecting the progressive appearance of synchronized epileptiform activities in neuronal ensembles upon Mg^2+^ washout (see examples in Figure 7). These results are consistent with those of previous electrophysiological and imaging studies [30,41,42]. Similar to earlier reports [30,43,44], we observed that slowly propagating waves of intense bioluminescence occurred in some slices (*n* = 2 out of 2 and *n* = 8 out of 14 GA-expressing and RedquorinQ159T-expressing slices, respectively) after prolonged perfusion with Mg^2+^-free solution (Figure 7, and Appendix A). These waves temporarily interrupted epileptiform activities (Figure 7) and are reminiscent of cortical spreading depression [45]. These results indicate that, like GA, RedquorinQ159T is a suitable bioluminescent Ca^2+^ sensor for the wide-field imaging of diverse neuronal network dynamics with a high signal-to-background ratio.

## 3. Discussion

### 3.1. Apoaequorin Mutations That Enhance Aequorin Ca^2+^ Sensitivity

Aequorin light emission rate/intensity increases by several orders of magnitude over the [0.1 µM–1 mM] Ca^2+^ concentration range [6]. This increase is determined by the interplay between three EF hands endowed with different Ca^2+^ affinities, with a low affinity EF1 domain and a high affinity domain comprising the EF2,3 cooperative pair [7]. In the present study, 44 apo-aequorin mutations were screened, of which 7 enhanced the Ca^2+^ sensitivity of aequorin. None of the mutations generated in or around EF1 enhanced aequorin Ca^2+^ sensitivity. In particular, mutations flanking the N26 Ca^2+^-binding residue, whose N26D substitution enhances aequorin Ca^2+^ sensitivity [7,35], either reduced (V25I) or did not affect (V25A, H27N, H27G) Ca^2+^ sensitivity. Conversely, the 7 mutations that enhanced aequorin Ca^2+^ sensitivity concern four Ca^2+^-binding residues located at equivalent positions in EF2 (N121, A123) and EF3 (S157, Q159), as well as a residue of the C-terminal tail (A179). Among these five residues, four (N121, A123, Q159, A179) bear the substitutions that have been used, alone or in dual combinations, to enhance Redquorin Ca^2+^ sensitivity. The functional impact of these mutations can be tentatively explained in light of aequorin crystal 3D structures (Ca^2+^-free aequorin: [10], PDB code 1EJ3; Ca^2+^-bound apo-aequorin: [11], PDB code 1SL8), in which N121, A123, Q159, and A179 lie within short distance of each other and participate in Ca^2+^ binding, EF2-EF3 interactions, and/or EF3-coelenterazine binding domain interactions (see Figure 8).

The N121 EF2 residue interacts via its polar side chain with both Ca^2+^ and the Q159 EF3 residue. High Ca^2+^ sensitivity of the N121D mutant (not observed for N121S) may primarily result from the negatively charged side chain of the D residue strengthening interaction with Ca^2+^, thus enhancing EF2 Ca^2+^ affinity. The A123 EF2 residue binds Ca^2+^ via the carbonyl of its peptide bond, and its short hydrophobic side chain points towards the Q159 side chain. As for the N121D mutation, the A123D substitution (but not A123S and A123T) may primarily enhance EF2 Ca^2+^ affinity by increasing its interaction with Ca^2+^. The Q159 EF3 residue binds Ca^2+^ via the carbonyl of its peptide bond, and its polar side chain interacts with that of the N121 residue. The replacement of Q159 by K (long side chain) does not change Ca^2+^ sensitivity, whereas Q159D, T, or G mutants (shorter or no side chain) presumably result in the loss of interaction with N121, thereby enhancing EF2 Ca^2+^ affinity by unleashing N121 interaction with Ca^2+^. The A179 residue points towards the beginning of EF3 and neighbors the C180 residue. The C180 residue interacts with the Q168 residue, which belongs to an α-helix comprising several coelenterazine-binding residues downstream of EF3 and interacts with the E164 Ca^2+^-binding residue of EF3 [35]. In contrast with A179 (short hydrophobic side chain), the mutant T179 (longer polar side chain) may interact with the EF3 residue and/or alter interaction between C180 and Q168 residues, thereby enhancing EF3 Ca^2+^ affinity and/or facilitating the transduction of EF3 occupancy by Ca^2+^ to light emission. It is noteworthy that the above hypotheses are compatible with observation of the additive effects of these mutations in dual combinations. Moreover, these hypotheses predict that the mutations also affect the interplay between aequorin EF hands, and thus not merely shift aequorin Ca^2+^ sensitivity curve, but additionally change its shape/slope, as indeed observed in the present study.

### 3.2. AequorinXS Mutations Primarily Enhance Redquorin Sensitivity to Low Ca^2+^

The Ca^2+^ sensitivity of Redquorin mutants was compared to those of GA, CitA, and wt-Redquorin, whose Ca^2+^ sensitivity curves are essentially parallel to that of wt-aequorin, but shifted towards lower (GA and CitA), or higher (wt-Redquorin) Ca^2+^ concentration. Our results show that AequorinXS mutations enhanced Redquorin Ca^2+^ sensitivity in all regards (submicromolar sensitivity, EC50, decay kinetics). However, this effect was more prominent at low than at high Ca^2+^ concentration. Indeed, all Redquorin mutants exhibiting higher sensitivity to submicromolar Ca^2+^ concentration than GA, and some higher than CitA. Conversely, the increase of light emission intensity with Ca^2+^ concentration was smaller for RedquorinXS mutants than for GA and CitA, as apparent in the Ca^2+^ sensitivity curves of all mutants crossing those of GA and/or CitA, and in the decay kinetics of all mutants being slower than those of GA and CitA at saturating Ca^2+^ concentration. These different slopes point to differences in mechanisms of Ca^2+^ sensitivity enhancement. While RedquorinXS mutations target EF2 and EF3 with likely alterations of the three EF interplay, fusion with GFP in GA and CitA results in a global enhancement of aequorin Ca^2+^ sensitivity that presumably leaves the three EF interplay unaffected. GA and CitA fusions mimic the natural, non-covalent association of GFP and aequorin in *Aequorea victoria* photocytes [2]. This suggests that the enhancement of the Ca^2+^ sensitivity of their aequorin moieties occurs through the GFP/aequorin interface optimized through co-evolution. In contrast, although high BRET efficiency attests to their close proximity in Redquorin [33], aequorin fusion with tdTomato does not enhance its Ca^2+^ sensitivity, suggesting that the aequorin-tdTomato interface may be engineered as an alternative strategy to enhance Redquorin Ca^2+^ sensitivity.

### 3.3. RedquorinXS Sensors Report Intracellular Ca^2+^ Signals with High Sensitivity in Cell Lines and in Brain Slice

Consistent with their enhanced Ca^2+^ sensitivity, the four tested RedquorinXS mutants surpassed Redquorin in sensing P2Y2 receptor activation by ATP upon expression in CHO cell lines. As expected from their respective Ca^2+^ sensitivity in vitro, the dose-response curves of RedquorinQ159D and Q159T mutants in the P2Y2 receptor in cellulo assay exhibited lower [ATP] detection limits than GA, and reached a maximum at higher [ATP] than GA. The performance of RedquorinQ159D and Q159T in the P2Y2 receptor assay was thus superior to that of GA, in terms of ATP detection sensitivity and of ATP concentration range reported. Moreover, the P2Y2 receptor assay was more robust (Z-factor) using RedquorinQ159T as reporter than using GA. Conversely, and at odds with their respective Ca^2+^ sensitivity in vitro, RedquorinQD+AT and QT+AD double mutants were less sensitive reporters of P2Y2 receptor activation than RedquorinQ159D/T single mutants and GA. Protein misfolding or instability in cellulo are unlikely to account for this discrepancy, since photoproteins used in vitro were also produced in mammalian cells, and since thermostability was similar for all Redquorin mutants. Conversely, this discrepancy may result from differences between in vitro and in cellulo solutions. For example, Mg^2+^ at cytosolic concentration is a competitor of Ca^2+^ binding to EF hands, whose inhibitory effect on aequorin and GA is sensitive to EF mutation [23,46]. Cytosolic Mg^2+^ may thus preferentially inhibit Redquorin double over single mutants because of its enhanced binding to EF2 (Q159T+A123D mutant) or EF3 (Q159D+A179T), leading to lower in cellulo sensitivity of Redquorin double mutants. Our results thus indicate that the RedquorinXS Q159T mutant is the best suited aequorin-based red sensor for Gq-coupled receptor assays, but also for the simultaneous, multicolor monitoring of cellular signaling in combination with bioluminescent sensors for diverse intracellular signals emitting at various wavelengths [47,48,49,50,51,52].

An earlier study showed that the small proportion of red light emitted by the fusion of aequorin with red fluorescent protein mRFP can be detected through the mouse skull [31]. This suggests that Redquorin mutants, whose light emission is optimally shifted to red, may be used for transcranial monitoring of brain activity, pending their ability to report the activities of neuronal ensembles. We thus tested the mutant showing highest in cellulo performance, RedquorinQ159T, for the imaging of neuronal network activity following viral transfer in slices of mouse neocortex, using GA as an established reference [30]. Both sensors were expressed in cortical pyramidal neurons, exhibited a low basal signal in the absence of cortical stimulation, and similarly reported synchronized activities of neuronal ensembles elicited by the removal of extracellular Mg^2+^ via the emission of large bioluminescence peaks. These results do not allow for quantifying the relative sensitivity of the two Ca^2+^ sensors because of the high variability of neuronal network activity patterns evoked by Mg^2+^ washout [30,41,42]. Nonetheless, they demonstrate that RedquorinQ159T efficiently reports neuronal network dynamics in real-time upon wide-field bioluminescence imaging with a high signal-to-background ratio. Mg^2+^ removal also elicited waves of intense bioluminescence, which were observed for both Ca^2+^ sensors, propagated slowly across the cortical slice, and resembled cortical spreading depression. Spreading depression consists in a transient wave of near-complete neuronal and glial depolarization associated with large intracellular Ca^2+^ increases and is a pathophysiological correlate of migraine that also occurs following traumatic brain injury or focal ischemia [45]. Our results thus indicate that the RedquorinXS Q159T mutant is the best suited aequorin-based red sensor for the monitoring of normal and pathological neuronal network activities.

## 4. Materials and Methods

### 4.1. Site-Directed Mutagenesis of Apo-Aequorin and Redquorin

The construct of wt-apo-aequorin (PDB ID: 1EJ3; SEQ ID NO:1), and the fusion proteins wt-redquorin (wt-Redq) and citrine-aequorin (CitA), subcloned in pTriEx vector, were generated and obtained as a generous gift from Prof. Juan Llopis [32,33]. The fusion construct GFP-aequorin (GA) was the one generated in the lab of Prof. Philippe Brûlet [24]. The selected single and double site mutations were obtained using mutagenic oligonucleotide primers applied on wt-apo-aequorin or redquorin constructs using the single- and multiple-site QuikChange XL Site-Directed Mutagenesis kit according to the manufacturer’s instructions (Catalog #200516, Agilent Technologies, Les Ulis, France). The DNA sequence of all obtained mutants was verified by Sanger sequencing.

### 4.2. Production and Purification of Apophotoproteins from Mammalian HEK Cells

HEK-293 cells (kind gift from Dr. Helene Faure) were cultured in Dulbecco’s Modified Eagle’s Medium (Lonza, Levallois, France) supplemented with 2 mM L-glutamine, 10% heat-inactivated fetal bovine serum (FBS) and 100 U/mL of penicillin/streptomycin (Lonza). The cells were routinely maintained at humidified atmosphere of 37 °C and 5% CO_2_. For transfection with foreign DNA vector, the cells were seeded at a density of 6 × 10^5^/cm^2^ and transfected the day after with Lipofectamine 2000 (Life technologies SAS, Villebon-sur-Yvette, France) according to the manufacturer’s recommendations. One day post-transfection, HEK-293 cells (2–3 × 10^6^ cells) transiently expressing the apophotoproteins were rinsed twice with phosphate-buffered saline (PBS) and collected using a cell scraper, under ice-cold conditions. The cells were harvested at 500g for 5 min and then resuspended and washed once in cold PBS buffer. Subsequently, the cells were lysed using a hypo-osmotic buffer composed of 20 mM Tris-HCl (pH 7.5), 10 mM EGTA, and 5 mM β-mercaptoethanol, prepared in Milli-Q/H_2_O and supplemented with a protease inhibitor cocktail (complete-Mini, EDTA-Free, Roche, Basel, Switzerland). The cell membranes were broken by two freeze-thaw cycles, followed by few passages through a 25-gauge needle. The resulting lysates samples were then centrifuged at 13,000× *g* for 20 min to remove cell debris and unbroken cells. The 500 μL supernatant containing the apophotoprotein was recovered and passed through molecular weight cut-offs (*AmiconUltra* MWCO, 10K and 50K, from Millipore SAS, Molsheim, France). Columns of 10 kDa were used for apo-aequorin mutant proteins and 50 kDa for redquorin mutant proteins. This step allowed for higher purity and buffer exchange by eliminating unwanted proteins, salts, and different compounds in the eluted protein samples. The samples were concentrated from approximately 500 μL to 20 μL volumes for buffer exchange. The 20 μL concentrated samples were resuspended in 500 μL volume of the desired buffer and this step was repeated twice to ensure high fidelity wash and buffer exchange. The concentrated samples were stored at 4 °C for reconstitution in future assays.

### 4.3. Protein Expression and Purification from E. coli

Some experiments of affinity, emission lifetime, and thermostability were repeated on highly purified proteins and gave similar results (Redq/Q159D; Redq/QD+AT, Redq/Q159T, Redq/QT+AT, and Redq/QT+AD). Consult more detailed protocol in the following references [32,33]. Briefly, the expression of His-tagged photoproteins was carried out in *E. coli* using pTriEx-4 plasmid system. Bacterial cells expressing the photoproteins were then lysed and photoproteins were column purified using Ni-NTA HisBind resin (Merck Millipore, Burlington, MA, USA). An additional purification step was performed using molecular weight cut-offs (*AmiconUltra* MWCO, 50K, from Millipore) as described in the previous section of cell culture. Protein quantification of the protein samples was performed on Nanodrop machine (ThermoFisher scientific, Waltham, MA, USA), by using a combination of absorbance either at 280 nm or at wavelength that corresponds to the maximum extinction coefficient of the fluorescent protein tdTomato in Redq (554 nm). The pure samples were stored at 4 °C for reconstitution in future assays.

### 4.4. Aequorin Reconstitution for In Vitro Assays

The purified and concentrated samples of apo-aequorin and redquorin mutant proteins were buffer-exchanged to 50 mM Tris–HCl (pH 7.5), 150 mM NaCl, 10 mM EGTA, 5 mM β-mercaptoethanol, supplemented with 5 μM CLZ-f or CLZ-native and incubated overnight at 4 °C, in the dark. The next day, samples were passed through molecular weight cut-offs (*AmiconUltra* MWCO, 10K and 50K, from Millipore), then resuspended in Zero-Ca^2+^ buffer (30 mM MOPS, 100 mM KCl and 10 mM EGTA, pH 7.2) for calcium affinity, emission decay kinetics, and spectral analyses assays.

### 4.5. Functional Analysis of Aequorin Mutants and Fusions In Vitro

*Ca^2+^ sensitivity curves.* Purified and reconstituted samples with *CLZ-f* or *CLZ-native* already buffered-exchanged to zero-calcium buffer ensured to minimize calcium contamination that could interfere with the measurements. A fifteen microliter aliquot of photoprotein samples (concentration range, 50 to 90 ng/μL) was placed in a home-made luminometer and mixed with 300 μL solution containing the desired concentration of free-Ca^2+^. The solutions of EGTA-buffered with varying free-Ca^2+^ prepared according to the manufacturer instruction for the kit (Molecular Probes, Invitrogen). The integration of luminescence emission per second was measured for 20 s (background) before injecting solution with known [free-Ca^2+^] and lasted until stable or decaying signal (at final-phase) was obtained. At this point, 400 μL of a saturating calcium solution (30 mM MOPS, 100 mM CaCl_2_, pH 7.2) was rapidly injected (less than 30 ms) and light integration continued until all aequorin had been consumed and light intensity had gone back to background signal. Luminescence intensity (*L*) at different [Ca^2+^] was taken as the response peak value in case of decaying signal or as the stable value in case of stable emission. *Lmax* was measured by integrating all remaining luminescence signal from that moment (*L* value) to the end of the experiment. The *EC50* values were extracted from a sigmoidal dose-response fit (variable slope) using Prism software ( GraphPad, San Diego, CA, USA).

*Emission half-life*. For assessing the decay kinetics of the mutant proteins of aequorin and redquorin, as well as citrine-aequorin (CitA) and GFP-aequorin (GA), the emission half-life (*t_1/2_*) was calculated from the curve with monoexponential decay fit. The recording of the luminescence signal (sampling interval of 30 ms) started before the fast injection of 200 μL of saturating calcium solution (50mM Tris–HCl, 100 mM CaCl_2_, pH 7.5) into fifteen microliter aliquot of protein samples. This resulted in a peak response with complete aequorin consumption and emission decay in less than 8 s for all samples.

*Spectral measurement and analysis.* A fifteen microliter aliquot of photoprotein samples was brought in contact with 200 μL saturating calcium solution (50 mM Tris–HCl, 100 mM CaCl_2_, pH 7.5) in a PCR tube to induce light emission. Emitted photons were collected via an optic fiber, guided into a spectrometer (Specim, Oulu, Finland), and captured by an EM- CCD camera (DU-897 back illuminated, Andor, Belfast, Northern Ireland). Spectral calibration was performed using laser pointers (405 and 650 nm). This home-made setup, from the lab of Prof. Philippe Brûlet, allowed immediate and synchronous spectral analysis (no scanning) of emitted luminescence with 1-nm resolution [31].

*Thermostability.* The purified photoprotein samples were reconstituted with CLZ-f (as detailed earlier) and let to calibrate at room temperature (20–24 °C) for 20 min before taken the first measurement of total counts using saturating calcium solution (50 mM Tris–HCl, 100 mM CaCl_2_, pH 7.5). The samples were then incubated at two different temperatures for 30 min and let equilibrate at room temperature for 15 min before taken the second measurement of total counts in the sample. The relative bioluminescence activity at each temperature was calculated by the ratio 2^nd^ counts/1^st^ counts multiplied by 100.

### 4.6. Functional Analysis of Photoproteins in CHO Cell Lines

CHO cells stably expressing photoproteins. To produce stable CHO cell lines, the cells transfected with GA and Redquorin variant plasmids were seeded on six-well plates and grown in the same medium with addition of Geneticin sulfate (G418) (Thermo Fisher Scientific, Waltham, MA USA) at 1 mg/mL under the same conditions with replacement of the medium by fresh medium every 2 days. After 7–10 days of selection, CHO cells were transferred into a medium without antibiotic. To select clones with the highest bioluminescence activity, one run of limiting dilution was performed. Following G418 selection, CHO cells were seeded on 96-well plates (approximately 0.5 cells per well) and grown in DMEM/F12 medium supplemented with 10% fetal calf serum at 37 °C, 5% CO_2_ for up to 80–90% confluence. Before the bioluminescence measurements, the plates with monoclones of CHO cells expressing a given apophotoprotein were duplicated and grown up to 90–100% confluence. The medium was then removed and cells were loaded with 5 μM *CLZ-native* in Tyrode solution (130 mM NaCl, 5 mM KCl, 1 mM MgCl_2_, 2 mM CaCl_2_, 5 mM NaHCO_3_, 20 mM N-(2-hydroxyethyl) piperazine-N′-ethanesulfonic acid (HEPES), pH 7.4) for 3 h at room temperature. Then, bioluminescence in each well was measured with a Mithras LB 940 plate luminometer (Berthold Technologies GmbH, Bad Wildbad, Germany) at 23 °C by injection of 1% Triton X-100 in Tyrode solution for cell lysis and triggering light emission. Bioluminescence was measured by integrating the light signal for 10 s. The CHO cell clones with the highest bioluminescence activity were selected for this study.

*Assay of activation of endogenous P2Y2 receptor in CHO cells.* The CHO cell lines stably expressing each Redquorin variant or GA were used in assay of activation of endogenous P2Y2 receptor by ATP. The day before the measurements, the corresponding cells were seeded on 96-well plates with DMEM/F12 medium supplemented with 10% fetal calf serum and were grown up to 90–100% confluence at 37 °C with 5% CO_2_. Then, the medium was replaced by 100 μL per well of the coelenterazine solution described above, and cells were incubated at 23 °C for 3 h. ATP was injected to trigger an intracellular Ca^2+^ response in the cells and bioluminescence measurement started immediately. Bioluminescence was measured using a Mithras LB 940 plate luminometer, with light emission integrated over a 0.5 s time interval for 60 s.

### 4.7. Bioluminescence Imaging of Photoproteins in Mouse Neocortical Slices

*Preparation and production of recombinant viral vectors.* Recombinant Sindbis viruses were prepared and used to express Redquorin mutants and GA in neurons of brain slices as described [23,53]. The coding sequences of Redquorin mutants and GA were first subcloned in the plasmid pSinRep5 (Invitrogen) upstream to the polyA signal. The resulting pSinRep5 plasmids encoding the fusion protein sensors as well as the helper plasmid pDH26S (Invitrogen) were then submitted to in vitro transcription to prepare capped RNA using the Megascript SP6 kit (ThermoFisher scientific, Waltham, MA, USA). Next, BHK-21 (baby hamster kidney; CCL-10; ATCC) cells were electroporated with both sensors-encoding and helper viral RNAs, and maintained for 24 h at 37°C, 5% CO_2_ in DMEM containing 5% fetal calf serum. Recombinant pseudovirions were harvested by collecting the cell supernatant and were stored at −80 °C.

*Preparation and viral transduction of neocortical Slices*. All experiments were carried out in accordance with the guidelines published in the European Communities Council Directive of September 22, 2010 (2010/63/UE, project agreement # APAFIS#16198-2018071921137716v3, approved by ethical committee #5 on 02/19/2017). Parasagittal sections (250 µm-thick) of cerebral cortex were prepared from young C57BL/6J mice (10–14 postnatal days old) as described [23,53]. The slices were incubated at room temperature for 30 min in artificial cerebrospinal fluid (ACSF) containing (in mM): 126 NaCl, 2.5 KCl, 1.25 NaH_2_PO_4_, 2 CaCl_2_, 1 MgCl_2_, 26 NaHCO_3_, 20 glucose, 5 pyruvate, 1 kynurenic acid and oxygenated with a mixture of 95% O_2_/5% CO_2_. The slices were then transferred onto a Millicell CM membrane (Millipore) pre-equilibrated with culture medium (50% MEM, 50% HBSS, 6.5 g/L glucose, 10 U/mL penicillin, 10 µg/mL streptomycin). The transduction of the slices with Sindbis pseudovirions as DNA delivery vehicles was carried out as described [23,53]. The apophotoproteins in Redquorin mutant and GA were reconstituted overnight by supplementing the culture medium with 10 µM native CLZ. The next day, cortical slices were pre-incubated for at least 1 h in ACSF before being transferred to the imaging chamber where ACSF was continually perfused at a rate of 1–2 mL/min.

*Bioluminescence imaging*. Bioluminescence imaging was performed as described [30] using an intensified CCD video camera (ICCD225; Photek, St Leonards on Sea, United Kingdom) controlled by the IFS32 software (Photek), and mounted on an upright microscope (BX51WI, Olympus, Tokyo, Japan) equipped with water immersion objectives ×10 (NA = 0.3) and ×20 (NA = 0.95). Prior to bioluminescence imaging, fluorescence images of GA- or Redquorin-transduced slices were acquired in the “bright field” mode of the camera using GFP (Semrock, Rochester, NY, USA) or Cy3ET (Chroma, Bellows Falls, VT, USA) filter sets. Bioluminescence imaging was performed at room temperature in “bining slice” mode at a rate of 25 frames per second in complete darkness and without emission filter to maximize photon capture. Data were collected, stored, and visualized with the IFS32 software.

*Immunohistochemistry*. Slices were fixed overnight at 4 °C in 0.1 M phosphate-buffered saline (PBS, pH 7.4) containing 4% paraformaldehyde. Next, the slices were washed with PBS 3 × 15 min and permeabilized with 0.2% gelatin/0.25% triton X-100 in 0.1 M PBS (GT-PBS) for 1 h. The slices were then incubated overnight with primary antibody in GT-PBS. Primary antibodies used were rabbit anti-Satb2 (1/1000; Abcam, ab34735), chicken anti-GFP antibody (GFP-1020, 1/2000; Aves Labs, Tigard, OR, USA) and rat anti-RFP (5F8, 1/1000; Chromotek, Planegg, Germany). After 3 × 10 min washes in GT-PBS, the slices were incubated for 2 h with Alexafluor 488-conjugated goat anti-rabbit IgG, and Alexafluor 555-conjugated goat anti rat IgG (1/1000; Life Technologies, A-11034 and A-21434 respectively). After at least three washes in 0.1 M PBS, slices were mounted in fluoromount-G (Clinisciences, Nanterre, France). All steps were performed at room temperature except the incubation with primary antibodies that was performed at 4 °C. Images were acquired with an upright Leica DM 6000 SP5 confocal laser scanning microscope, and then analyzed using the cell counter plugin of ImageJ 1.32 software (National Institute of Health, Bethesda, MD, USA). Counting was performed upon visual inspection of confocal images and no procedure was implemented to reduce count bias.

## 5. Conclusions

RedquorinXS-Q159T is an optimized red light-emitting Ca^2+^ sensor suitable for the monitoring of intracellular signaling in a variety of screening and basic research applications in cells and tissues.

## Figures and Tables

**Figure 1 ijms-21-07846-f001:**
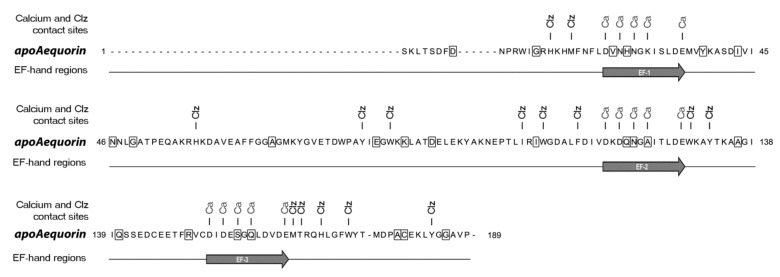
Apoaequorin amino acid sequence highlighting the mutated sites screened for enhanced Ca^2+^ sensitivity. The apo-aequorin sequence from *Aequorea victoria (PDB code: 1EJ3)* is 198 amino acids long. Random, non-conservative substitution was performed on boxed amino acids in order to alter aequorin Ca^2+^ affinity. Amino acids involved in coelenterazine (Clz) and Ca^2+^ (Ca) binding are indicated.

**Figure 2 ijms-21-07846-f002:**
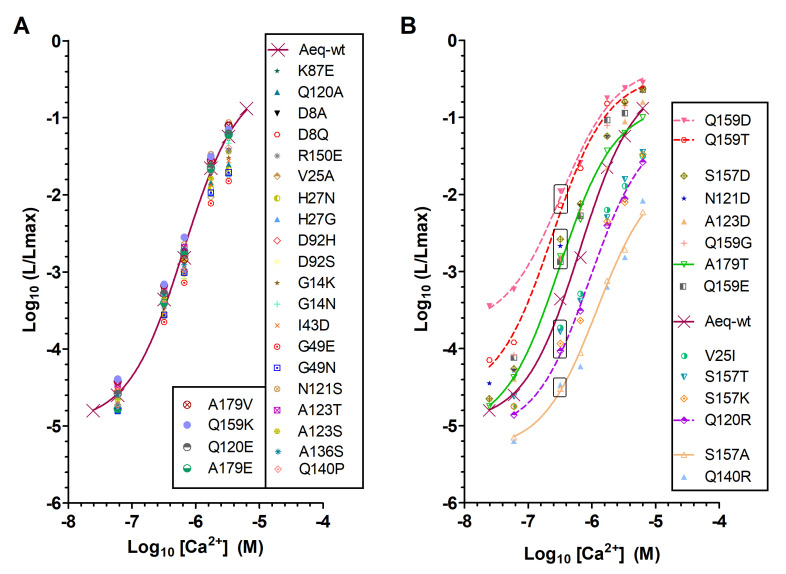
Ca^2+^ sensitivity of aequorin mutants. The relationship between Ca^2+^ concentration ([Ca^2+^]) and fractional bioluminescence intensity (L/Lmax) is displayed in Log values on a linear scale. L and Lmax stand for the peak luminescence intensity at a given [Ca^2+^] and the total luminescence intensity at saturating [Ca^2+^] for the same sample, respectively. Aequorin and its mutants were reconstituted with CLZ-f. For wt-aequorin (Aeq-wt), a sigmoidal curve fit is traced for comparison. R square of the fit goodness was higher than 0.994. (**A**) Amino acid mutations that resulted in no significant change in affinity, n was at least 2. (**B**) Amino acid mutations that increased or decreased Ca^2+^ sensitivity, n was at least 3. At pCa 6.5, and relative to wt-aequorin, four mutant groups (boxed) have been identified with low (e.g., Q140R), medium-low (e.g., S157T), medium-high (e.g., A123D) or high (e.g., Q159D) sensitivity to Ca^2+^.

**Figure 3 ijms-21-07846-f003:**
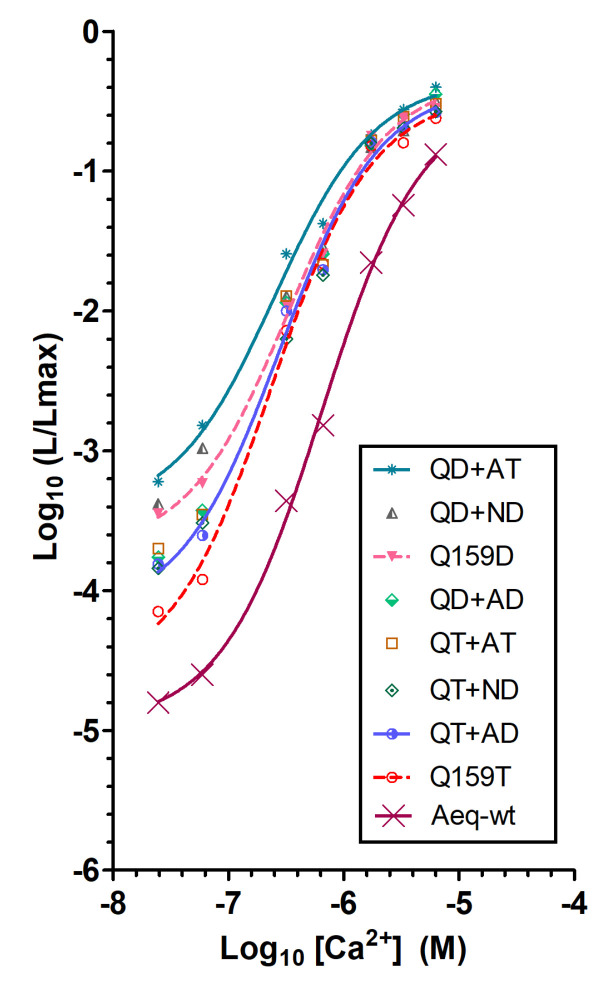
Ca^2+^ sensitivity of aequorin double mutants. The relationship between Ca^2+^ concentration ([Ca^2+^]) and fractional bioluminescence intensity (L/Lmax) is displayed in Log values on a linear scale. Aequorin reconstitution was with CLZ-f. The graph shows Ca^2+^ sensitivity of aequorin double mutants combining single mutations that resulted in Ca^2+^ extra-sensitivity (AequorinXS), n was at least 4. For wt-aequorin, a sigmoidal curve fit is traced for comparison. R square of the fit goodness was higher than 0.994. The following abbreviations are used for clarity: QD+AT (Q159D+A179T), QD+ND (Q159D+N121D), QD+AD (Q159D+A123D), QT+AT (Q159T+A179T), QT+ND (Q159T+ N121D), QT+AD (Q159T+ A123D).

**Figure 4 ijms-21-07846-f004:**
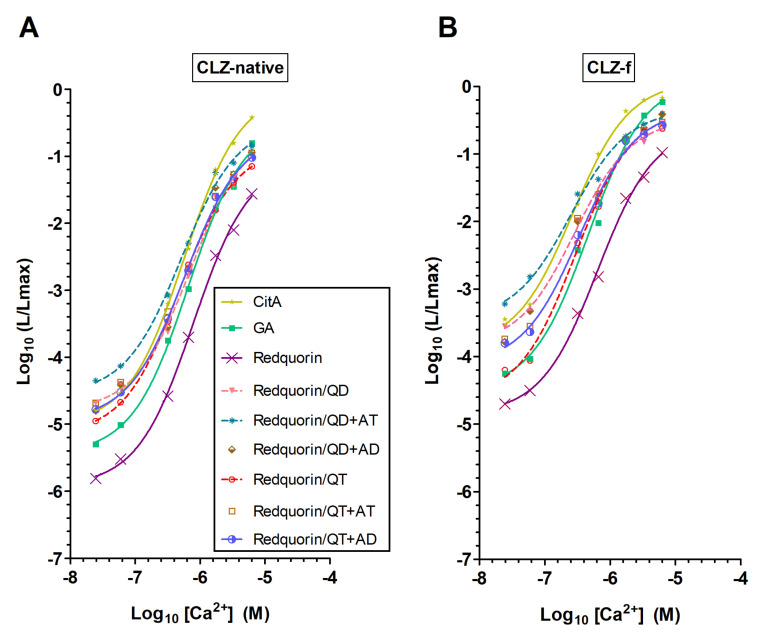
Ca^2+^ sensitivity of Redquorin and its mutants relative to GFP-aequorin (GA) and CitA. The relationship between Ca^2+^ concentration ([Ca^2+^]) and fractional bioluminescence intensity (L/Lmax) is displayed in Log values on a linear scale (*n* was at least 4). Aequorin moieties of the fusion proteins were reconstituted with native CLZ (**A**) or CLZ-f (**B**). For Redquorin (Rdq), a sigmoidal curve fit is traced for comparison. R square of the fit goodness was higher than 0.994.

**Figure 5 ijms-21-07846-f005:**
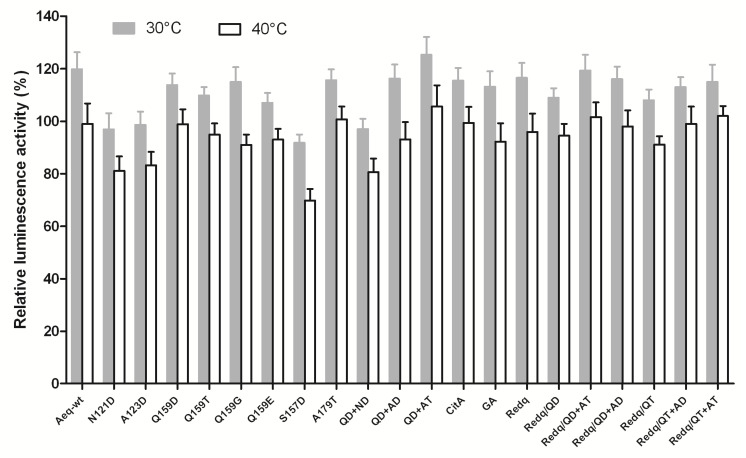
Thermostability of aequorin mutants, GA, CitA, redquorin (Redq) and Redq mutants. Photoproteins were purified and reconstituted with CLZ-f. The samples were incubated at room temperature (20–24 °C) for 15 min before measuring total luminescence on a first aliquot using saturating Ca^2+^ solution. Samples were then incubated at indicated temperatures for 30 min before measuring total luminescence on a second aliquot. The relative bioluminescence activity at each temperature was calculated by the ratio 2^nd^ counts (at target T °C)/1^st^ counts (at initial room T °C) multiplied by 100. Bars represent mean *± SD*, with n between 3 and 5.

**Figure 6 ijms-21-07846-f006:**
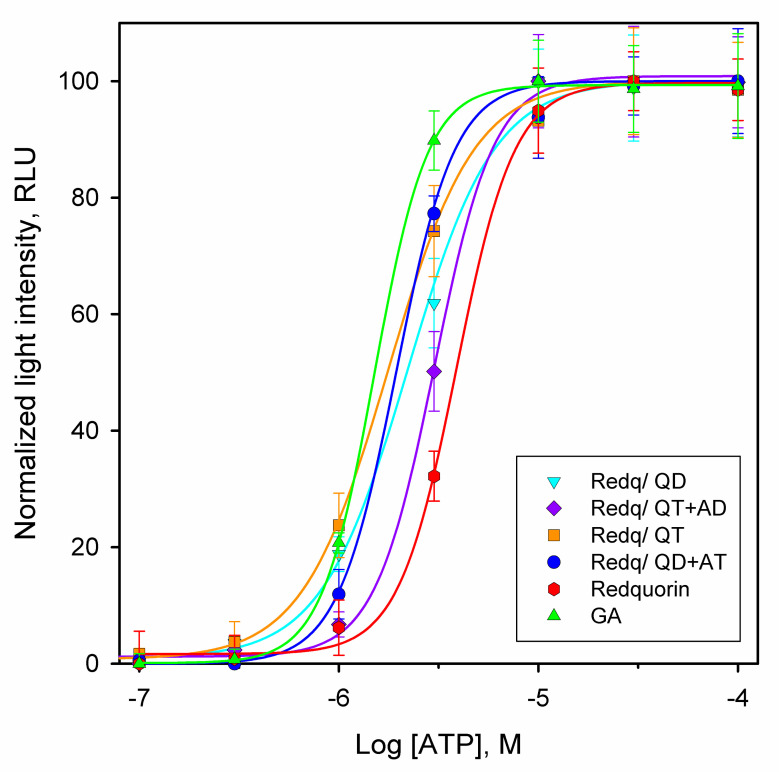
Dose–response curves for activation of endogenous P2Y2 receptor with ATP in CHO cells stably expressing the indicated Ca^2+^ sensor variants. The light intensity was measured by integration of the bioluminescence signal for 60 s. Each point is the mean ± SD of eight experiments.

**Figure 7 ijms-21-07846-f007:**
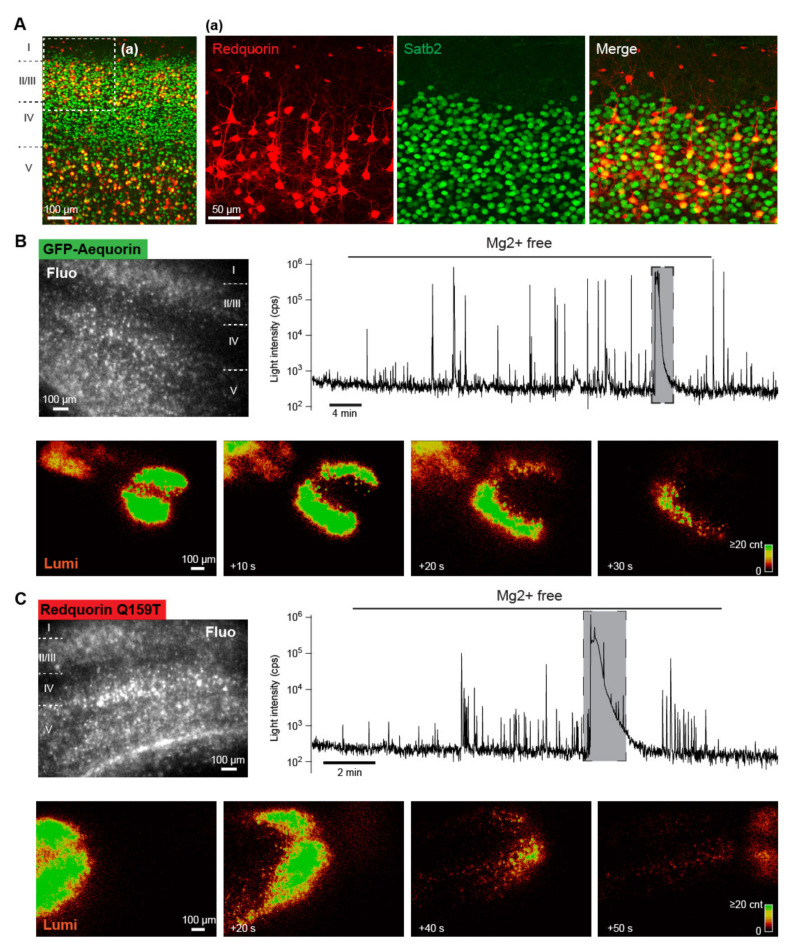
Ca^2+^ imaging of mouse neocortical network activities with GA and RedquorinQ159T. (**A**) Confocal reconstruction of a neocortical slice expressing RedquorinQ159T after viral gene transfer. Redquorin-expressing cells were preferentially localized in layers II/III and V, and most exhibited the typical morphology of pyramidal neurons and were immunopositive for the marker of excitatory neurons Satb2. (**B**,**C**) ***Upper***: Fluorescence images (Fluo) of neocortical slices expressing the indicated Ca^2+^ sensor, and traces corresponding to whole-field bioluminescence recordings of the same slices. Note the logarithmic scale of light intensity. ***Lower***: Bioluminescence images of the same slices showing intense bioluminescence waves corresponding to the large peaks on whole-field bioluminescence traces indicated by shadowed boxes. Note the slow propagation of the waves.

**Figure 8 ijms-21-07846-f008:**
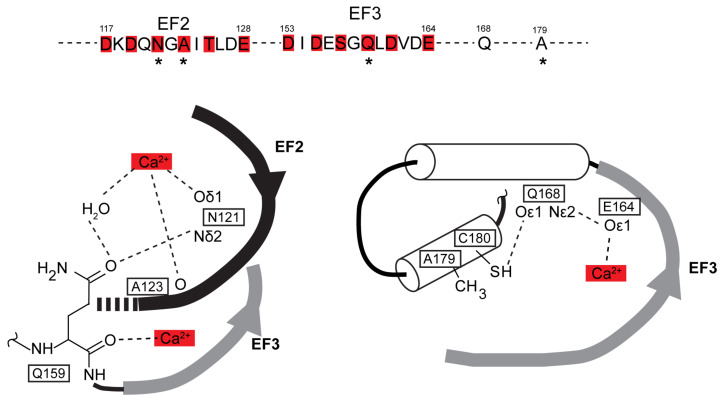
Local environment of amino acid residues bearing AequorinXS mutations N121D, A123D, Q159D/T, and A179T. ***Upper***: position of N121, A123, Q159, and A179 (asterisks) in the primary sequence of aequorin, red boxes denote Ca^2+^ binding residues. ***Lower***: local environment of these residues in the crystals of Ca^2+^-free aequorin (PDB code 1EJ3) and Ca^2+^-bound apo-aequorin (PDB code 1SL8). Note in left panel that EF2 residues N121 and A123 participate in Ca^2+^ binding, and that EF3 residue Q159 interacts with N121. Note in right panel that the A179 residue points via its side chain towards the beginning of EF3. Its neighbor C180 interacts with the Q168 residue, which belongs to an α-helix involved in coelenterazine binding and interacts with the EF3 Ca^2+^-binding residue E164.

**Table 1 ijms-21-07846-t001:** Properties of aequorin mutants with increased Ca^2+^ sensitivity.

	Mutation	Ca^2+^ Sensitivity, *EC50* (nM) ^a^	Relative Intensity ^a^		Decay Kinetics, *t_1/2_* (ms) ^b^	Emission Peak Wavelength (nm)
			at pCa 6.5	at pCa 7.2		
	Aeq-wt	659 ± 23	1	1	906 ± 53	476
**Single mutants with medium-high to high Ca^2+^ sensitivity**	**Q159D**	330 ± 8	25.1	23.4	794 ± 42	478
	**Q159T**	310 ± 13	16.6	4.8	866 ± 23	477
	**S157D**	322 ± 20	6.0	2.2	835 ± 47	475
	**N121D**	400 ± 14	4.9	2.1	957 ± 30	477
	**A123D**	374 ± 9	3.6	1.6	862 ± 48	476
	**Q159G**	492 ± 10	3.3	3.2	830 ± 37	478
	**A179T**	330 ± 19	3.0	1.3	779 ± 33	477
**Double mutants with high Ca^2+^ sensitivity**	**QD+AT**	216 ± 26	58.5	60.8	750 ± 50	478
	**QD+ND**	295 ± 17	29.0	41.7	738 ± 28	477
	**QD+AD**	271 ± 15	26.2	15.0	612 ± 26	475
	**QT+AT**	281 ± 25	29.0	14.0	650 ± 68	478
	**QT+ND**	337 ± 10	14.4	12.2	680 ± 55	478
	**QT+AD**	279 ± 16	22.8	10.0	620 ± 33	476

Aequorin and its mutants were reconstituted with CLZ-f. Ca^2+^ sensitivity (represented in *EC50*) was calculated from sigmoidal curve fit; n was at least 3. Values are displayed as mean ± SD. Values for Aeq-wt are in bold as a reference. Luminescence relative intensity (compared to Aeq-wt) was deduced from data points of L/Lmax for two Ca^2+^ concentrations, pCa 6.5 and pCa 7.2. ***a***, the R square of the goodness of the curve fit was higher than 0,994. The decay kinetics was assayed by fast-flow injection of a saturating Ca^2+^ solution and recording of the luminescence signal decay at a sampling interval of 30ms. Mean values ± SD were calculated from a monoexponential decay curve fit. ***b***, the R square of the goodness of the decay curve fit was higher than 0,999. Aequorin double mutants: QD+AT (Q159D+A179T); QD+ND (Q159D+N121D); QD+AD (Q159D+A123D); QT+AT (Q159T+A179T); QT+ND (Q159T+N121D); QT+AD (Q159T+A123D).

**Table 2 ijms-21-07846-t002:** Properties of Redquorin and its mutants relative to GA and CitA.

	Mutation	Ca^2+^ Sensitivity, *EC50* (nM) ^a^	Relative Intensity ^a^		Decay Kinetics, *t_1/2_* (ms) ^b^	Emission Peak Wavelength (nm)
			at pCa 6,5	at pCa 7,2		
**CLZ-native**	**Redq**	859 ± 45	1.0	1.0	1 203 ± 70	582
	**Redq/Q159D**	680 ± 33	9.1	13.2	980 ± 66	582
	**Redq/QD+AT**	515 ± 20	32.1	24.5	913 ± 39	582
	**Redq/QD+AD**	621 ± 44	10.8	12.8	1 120 ± 46	582
	**Redq/Q159T**	478 ± 39	13.0	7.1	1 103 ± 50	582
	**Redq/QT+AT**	605 ± 16	18.3	14.1	990 ± 87	582
	**Redq/QT+AD**	577 ± 33	14.4	10.0	950 ± 56	582
	**CitA**	573 ± 45	24	9.5	794 ± 46	529
	**GA**	630 ± 29	6.8	3.2	852 ± 64	509
**CLZ-f**	**Redq**	600 ± 35	1.0	1.0	880 ± 68	582
	**Redq/Q159D**	290 ± 18	23.9	14.9	770 ± 88	582
	**Redq/QD+AT**	252 ± 40	58.5	48.3	740 ± 55	582
	**Redq/QD+AD**	300 ± 26	22.8	15.0	750 ± 66	582
	**Redq/Q159T**	296 ± 13	11.0	2.8	705 ± 36	582
	**Redq/QT+AT**	284 ± 30	25.6	9.0	810 ± 61	582
	**Redq/QT+AD**	336 ± 11	14.4	7.4	670 ± 50	582
	**CitA**	266 ± 18	41.1	18.7	680 ± 72	529
	**GA**	463 ± 46	8.6	2.9	700 ± 48	509

Aequorin moieties of Redq, Redq mutants², GA, and CitA were reconstituted either with CLZ-native (top section of the table) or CLZ-f (bottom section of the table). For more details on Ca^2+^ sensitivity (represented in EC50), luminescence relative intensity and decay kinetics, refer to Table 1. Values for Redq are in bold as a reference. ***a***, the R square of the goodness of the curve fit was higher than 0.995. *n was* at least 3. ***b***, the R square of the goodness of the decay curve fit was higher than 0.999. *n* was at least 3.

**Table 3 ijms-21-07846-t003:** Performance of Redquorin mutants in cellular assay for P2Y2 receptor activation.

Ca^2+^ Sensor Variant	[ATP] CL (nM)	EC_50_ (µM)	*Z-Factor*
Redq/Q159T	310	1.7 ± 0.2	0.76
Redq/Q159D	320	2.2 ± 0.1	0.68
Redq/QD+AT	548	2.0 ± 0.2	0.67
Redq/QT+AD	1025	3.0 ± 0.4	0.82
Redq	2033	4.0 ± 0.3	0.56
GA	640	1.5 ± 0.2	0.69

CL: concentration limit of [ATP] is the smallest amount of ATP capable of generating measurable Ca^2+^ response by the sensor compared to background signal. It was estimated as the [ATP] that corresponds to three times the standard deviation of the intensity of the background signal; EC50: half-maximal effective concentration.

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
