# Peer review of "RedquorinXS Mutants with Enhanced Calcium Sensitivity and Bioluminescence Output Efficiently Report Cellular and Neuronal Network Activities"

_ijms, 2020, doi:10.3390/ijms21217846_

Round 1

Reviewer 1 Report

This manuscript by Bakayan et al details the development of redshifted aequorin mutants with improved calcium sensitivity for use as improved FRET-based reporters. The authors provide a good background of the state of the technology and provide sufficient details to understand their conclusions and support their work. In some instances, the manuscript is unnecessarily verbose, but scientifically it appears to be sound. The only exception is that one conclusion – presented in the Discussion section – is not well supported and should be modified. Overall, I suggest only minor revisions prior to acceptance. Specific issues are detailed below:

Major Issues – None

Minor Issues

  1. In the first paragraph of the Introduction section (Lines 46 – 64) the authors refer to Photoproteins as Ca2+ -dependent bioluminescent proteins. While this is true of aequorin, it is not true of all photoproteins (i.e., bacterial luciferase or firefly luciferase). The authors should amend the wording to refer only to the subset of photoproteins they are referencing (i.e, Ca2+ -dependent photoproteins).
  2. Line 60 should read “… is emitted at a slow rate…
  3. The authors should consider breaking the upper and lower halves of Table 1 into two separate tables for easier reference.
  4. The authors should consider moving Table 3 to supplemental material since it is not significantly different than what is presented in Figure 5.
  5. Figure 6 will be very difficult to interpret if printed in black and white. The authors should use different symbols, not just different colors, to represent the different proteins. Similarly, they should include the legend in the figure and not just in the text of the legend.
  6. Line 294 should read … number of compounds are evaluated…
  7. Line 311 should read … used for real-time monitoring of intracellular…
  8. The first paragraph of the Discussion section (Lines 356 – 373) is a simple summary and this information is mostly repeated in the following paragraphs anyway. It should be removed in the interest of brevity.
  9. Line 437 should read … reporter activation by lover ATP concentration limits than GA…
  10. In the final paragraph of the Discussion section (Lines 457 – 477) The authors state that the RedquorinXS Q159T mutant is an improved reporter for transcranial monitoring. This is not supported by the evidence provided. Although the mutant is shown to perform similarly, it has not been vetted in vivo under these circumstances. Unless additional experiments are performed to support this claim, it should be removed.

Reviewer 2 Report

I am computational chemist with an interest in fluorescent proteins. Although interested in the use of FPs and in aequorin this is not my area expertise. Having expressed my caution in my expertise I would like to say this is one of the best papers I have reviewed in a while. The results aren't earth shattering Nature/Science worthy, but they are solid well thought out, well described extensions of existing work in an important area.

It has a good introduction that is comprehensive yet concise. Good use of tables and figures. Text and research clear, precise  logical extension of redquorin work  Would be interested in knowing how these Ca sensor compare to red GECIs etc.

In discussion 3.1 I would have liked a figure of aequorin annotated with mutants being discussed.
